# Maternal Mental Illness Is Associated with Adverse Neonate Outcomes: An Analysis of Inpatient Data

**DOI:** 10.3390/ijerph16214135

**Published:** 2019-10-27

**Authors:** Abigail Lopez, Monideepa B. Becerra, Benjamin J. Becerra

**Affiliations:** 1Department of Health Science and Human Ecology, California State University, San Bernardino, CA 92407, USA; lopea439@coyote.csusb.edu; 2School of Allied Health Professions, Loma Linda University, Loma Linda, CA 92350, USA; bbecerra@llu.edu

**Keywords:** postpartum, mental health, mental illness, maternal health, child development

## Abstract

*Objective*: Addressing mental illness and associated outcomes is a major public health priority in the United States. In this study, our goal was to assess the role of maternal mental illness and its association to poor fetal growth and preterm delivery in one of the most socioeconomically disadvantaged areas of California. *Methods*: Data were obtained from the public database of California inpatient data from the Office of Statewide Health Planning and Development (OSHPD). OSHPD provides de-identified data on all inpatient department visits within California, to ensure confidentially of patients. Each variable was dichotomized into a binary variable of presence or absence of diagnosis status. The primary independent variable was clinical diagnosis of any mental illness. The dependent variables were pregnancy birth outcomes defined as poor fetal growth and preterm delivery. We specifically focused on inland Southern California due to its higher socioeconomic burden and poor maternal–child outcomes. *Results*: In the inland Southern California area, which is generally a geographic location with high poverty, maternal mental illness was associated with 79% higher odds of poor fetal growth and 64% higher odds of preterm delivery. Increasing numbers of co-morbidities were also associated with poor fetal growth. On the other hand, being older, being on Medicaid or other insurance status, being non-Hispanic Black, as well as increasing co-morbidities were associated with increased odds of preterm delivery. *Conclusions*: The purpose of the study was to evaluate the immediate birth outcomes associated with maternal mental illness. Given the empirical evidence of the study, addressing maternal mental health status is a key public health issue, especially in socioeconomically disadvantaged areas.

## 1. Introduction

The Centers for Disease Control and Prevention (CDC) define mental health as the state of wellbeing in which individuals can be productive in their everyday lives, can cope with their daily stresses, and can participate and contribute to their community [1]. On the other hand, mental illness represents various health conditions that are diagnosed by the alterations of mood, behavior, and ways of thinking. Only 17% of the United States population is considered to have an optimal mental health state, while 25% suffer from a mental illness [1]. In 2006, 36.2 million Americans paid for mental health services [2]. In the United States, $86 billion is spent on childbirth-related care every year [3], exemplifying the need for further research to find solutions to this ongoing dilemma. 

A major public health issue and research priority related to mental illness has been maternal mental health status. Pregnancy and postpartum are often culturally viewed as favorable outcomes for many women, but such a time is also characterized by adaptation to the process of learning and discovering new roles and responsibilities that come along with motherhood. In turn, this may lead to mixed emotional changes, including feeling a loss of control, stress, anxiety and/or depression. This can further put women at greater risk for any mental illness, which can also create mental disorders in the offspring [4].

There exists a plethora of research on maternal mental health status and offspring outcomes, though much of the empirical evidence focuses on long-term outcomes. For example, Cheng and colleagues [5] demonstrated that maternal health is associated with the child’s decreased overall physical health, constant tantrums, poor interaction skills, and complications in managing the child’s behaviors. The mother’s psychological stress during pregnancy is associated with the child’s temperament at 1 year of age, while anxiety in a mother predicted clinging behavior, frequent crying, and irritability in the child [6]. Studies also show that presence of a mental illness in a mother may influence a child’s emotional, cognitive, and behavioral attitude in addition to complications during birth, as well as a postpartum outcome of poor fetal growth [6]. Maternal depression in the postnatal period is associated with the child’s poor language development at 36 months [7]. 

Despite the aforementioned empirical evidence on postnatal mental health and child development, research on maternal mental health during pregnancy and resulting child outcome during birth is limited. As such, there is an imperative need to identify how maternal mental illness can impact immediate birth outcomes of both mother and child. Moreover, similar studies are common in developing nations. This may be attributable to the assumption that pregnancy and adverse birth outcomes may only be a developing world problem due to lack of proper care and resources; though, unfortunately, the same problem is occurring in developed nations [3]. United States data from 1990 to 2008 shows that while a majority of countries decreased their maternal mortality rates by 34%, the United States nearly doubled [3]. 

In addition, such outcomes are far worse in socioeconomically disadvantaged areas. For example, global studies have noted that living in poverty and having low education are associated with worse maternal–child outcomes, including preterm birth and general poor health status [8,9]. San Bernardino County, which is not only the largest geographic county in the United States but also has a substantially low socioeconomic status, has consistently reported high prevalence of poor maternal–child outcomes. For example, between 2014 and 2016, the infant mortality rate in the State of California was 4.3 per 1000, while in San Bernardino County, it was 5.9 per 1000, and in the surrounding Riverside County, it was 4.2 per 1000; together, these Counties form a majority of inland Southern California [10], colloquially known as the “Inland Empire”. Likewise, a report by the Southern California Association of Governments [11] notes that approximately 46% of the Inland Empire adults have a high school degree or less education. Likewise, 16.4% of adults and nearly 24% of children in the area live in poverty, compared to 15% and 20% in Southern California area, respectively. Furthermore, inflation-adjusted median income has remained stagnant in the region from 1990 to 2016, thus providing little scope of economic improvement for individuals. Despite having a high prevalence of poor maternal–child outcomes, limited studies have been conducted in this area, and thus, we aimed to focus on inland Southern California.

Furthermore, we aimed to focus on immediate birth outcomes, with specific emphasis on poor fetal growth and preterm delivery, associated with maternal mental illness. Poor fetal growth is known to be associated with later outcomes of increased rates of coronary heart disease, hypertension and type II diabetes [12]. Therefore, evaluating the key factors associated with such outcomes, especially related to maternal mental health, would provide for early preventative measures. Likewise, preterm birth has been shown to negatively impact health outcomes, including cognitive impairment, such as cerebral palsy [13]. As a result, this study further adds to the existing body of maternal–child health literature and provides scope for evidence-based improvement in maternal health care, a key objective of the Healthy People initiative [14].

## 2. Methods

### 2.1. Data Source and Population

To examine possible associations between maternal mental illness and immediate birth outcomes, inpatient 2008–2010 data from the California’s Office of Statewide Health Planning and Development (OSHPD) files were used for analysis. We did not combine with years post-2010 due to lack of key variables no longer included in the database. OSHPD is a public office that collects and distributes information on healthcare quality throughout California. They provide de-identified data on all hospital department visits to ensure the Health Insurance Portability and Accountability Act (HIPAA) policy and regulations are met for patient confidentiality. Further details of the OSHPD data can be found on their website at http://www.oshpd.ca.gov. In this study, the population included pregnant adult (aged 18 years or above) females with county of residence data within San Bernardino, Riverside, and Imperial counties of California, which we describe as inland Southern California. Males were excluded from the study; however, individuals with masked sex information were included if they met the definition of pregnancy used in the study. Pregnancy was defined as any individual with one of the following: International Classification of Diseases (ICD)-version 9, Clinical Modification (CM) diagnosis codes of: V27.* or 650; ICD-9-CM procedural codes of 720, 721, 722.1, 722.9, 723.1, 723.9, 724, 726, 725.1, 725.2, 725.3, 725.4, 727.1, 727.9, 728, 729, 732.2, 735.9, 736, 740, 741, 742, 744, 749.9; Medicare Severity-Diagnosis Related Groups (MS-DRG) codes derived from a Centers for Medicare and Medicaid (CMS) crosswalk file [15] for Diagnosis-related groups (DRG) DRG v24 Major Diagnostic Categories MDC of 14 for Pregnancy, Childbirth and the Puerperium: 765, 766, 774, 775, 767, 768. Individuals were excluded if they had abnormal (e.g., molar or ectopic pregnancies) or abortive pregnancy outcomes from ICD-9-CM diagnosis codes of: 630, 631, 632, 633.*, 634.*, 635.*, 636.*, 637.*, 638.*, 639.*; or procedural codes of: 690.1, 695.1, 749.1, 750.

### 2.2. Data Variables

The primary exposure variable in this study was presence of any mental illness using the Clinical Classification Software (CCS) codes of 657 (mood disorders), 658 (personality disorders), 659 (schizophrenia and other psychotic disorders), 662 (suicide and intentional self-inflicted injury), 656 impulse control disorders, 651 anxiety disorders, or 650 adjustment disorders. CCS codes were developed from the Agency for Healthcare Research and Quality (AHRQ) Healthcare Cost and Utilization Project (HCUP) and based on ICD-9-CM coding schemes [16]. The dependent variables in the study were birth outcomes defined as poor fetal growth (ICD-9-CM 656.50 and 656.51); and preterm delivery (ICD-9-CM 644.20 and 644.21). For reference, ICD-9-CM 656.50 is noted as “poor fetal growth, affecting management of mother, unspecified as to episode of care or not applicable”, while 656.51 is defined as “poor fetal growth, affecting management of mother, delivered, with or without mention of antepartum condition.” ICD-9-CM 644.20 and 644.21 are defined “early onset of delivery, unspecified as to episode of care or not applicable” and “early onset of delivery, delivered, with or without mention of antepartum condition,” respectively. 

The control variables of the study are age (18–64 years old), insurance type (Medicare, Medicaid, Private, or Other), race/ethnicity (defined as Hispanic, Non-Hispanic White, Non-Hispanic Black, and Non-Hispanic Asian/Pacific Islander), and the Charlson–Deyo Index. Medicare is a federally-sponsored and administered program through the Social Security Administration. It is a health insurance program for U.S. citizens and permanent residents who are 65 years and older, as well as those with specific disabilities and illnesses, such as end-stage renal failure [17]. Medicaid is federally and state-funded, and state-administered primarily for U.S. citizens and permanent residents who are low-income adults, children, pregnant women, as well as the elderly and those with disabilities [18]. In California, Medicaid is called MediCal. The Deyo modification of the Charlson comorbidity index provides a score for comorbidities for each discharged based on ICD-9-CM. This 17-item index has been shown to be a validated measure for administrative data [19]. 

### 2.3. Data Analysis

The first step in data analysis utilized the Chi-square test of association to assess the relationship between each of the study population characteristics and dependent variables. Next, logistic regression analyses were conducted to evaluate the association between maternal mental illness and immediate birth outcomes of poor fetal growth and preterm delivery. These analyses specifically utilized generalized estimating equations (GEE) models with a binary logit link to account for the hierarchical nature of discharges within hospitals and ZIP codes, which was treated by the model as nuisance factors. All statistical analyses were conducted using SAS v9.4 (SAS Institute Inc.; Cary, NC, USA) software and alpha less than 0.05 was used to denote significance. 

## 3. Results

Table 1 shows the prevalence of neonate outcomes by each population characteristic. Among women who reported any mental illness, prevalence of poor fetal growth was significantly higher as compared to those who did not (2.50% vs. 1.11%). Likewise, prevalence of preterm delivery was significantly higher among women with any mental illness compared to those without (12.93% vs. 6.69%). Other significant variables associated with higher prevalence of poor fetal growth and preterm birth were being 35–64 years compared to 18–34 years, being on Medicare, being non-Hispanic Black, and increasing co-morbidities. 

Table 2 demonstrates the odds of poor fetal growth given presence of any mental illness among mothers, after adjusting for control variables. Presence of mental illness was significantly associated with poor fetal growth and preterm delivery. Specifically, maternal mental illness was associated with 79% higher odds of poor fetal growth and 64% higher odds of preterm delivery. An increasing number of co-morbidities was also associated with poor fetal growth and preterm delivery. On the other hand, increased odds of preterm delivery were associated with being older, being on Medicaid or other insurance status, and being non-Hispanic Black.

## 4. Discussion

Our study evaluated the role of maternal mental health on postpartum outcomes, specifically poor fetal growth and preterm delivery. We chose these variables as they have been shown to negatively impact health outcomes [13,20]. Our study has several key findings: (1) the negative impact of maternal mental illness on both poor fetal growth and preterm birth, (2) the increased burden shared among non-Hispanic Blacks with preterm delivery as compared to non-Hispanic White mothers, and (3) number of co-morbidities were significantly associated with adverse birth outcomes. 

The association between maternal mental illness to that of poor fetal growth and preterm delivery warrants discussion. Previously, Hoffman and Hatch [21] evaluated 666 pregnant women to assess the putative association between depression symptoms during pregnancy and impact on fetal growth. While there was no overall relationship noted, in a smaller subset (*n* = 222), the authors found an association between increasing units on a standardized depression scale to that of lower birth weight. Specifically, the authors found that for each unit increase in the depressive symptom score at 28 weeks of gestation, there was a significant association to birth weight reduction (9.1 g). Likewise, in a meta-analysis on evaluating the role of depression, Grote et al. [22] noted that depression while pregnant was related to increased risk of preterm delivery as well as low birth weight. While such studies highlight the putative role of mental health on birth outcomes, our study does not rely on depression symptoms, which can be subjective due to self-reported nature. Instead, our study focuses on diagnosed mental illness and thus provides substantial evidence on the importance of incorporating mental health care as part of routine care to alleviate the long-term burden of mental illness on maternal–child health. 

Several other key findings in our study warrant further discussion. For example, we also noted that non-Hispanic Black mothers had higher odds of having preterm delivery. The current national data have already highlighted the disproportionate rate of health disparities shared by non-Hispanic Black mothers, and our study notes the imperative need to increase targeted prevention among such a population. For example, there is support in the literature for the use of faith-based initiatives to reach the African American/Black population [23], who may often defer routine medical care due to stigmatization. 

Additionally, in administrative data, “other” insurance is often self-pay, which is indicative of lack of insurance. Similarly, those on MediCal, California’s version of Medicaid, are likely to be of lower socioeconomic status. Given that delivery is often based on healthcare resources, it could likely explain why we only noted an association with preterm delivery but not growth. Regardless, given the higher burden noted in such populations, increased reach out is needed. For example, low-income populations are less likely to routinely use the healthcare system, and thus, home visitations during pregnancy and incorporation of community health workers [24,25,26] may aid in bridging such a gap and ensuring that those of low-income status may receive early preventative care to reduce the burden of maternal mental illness on preterm delivery.

Furthermore, we speculate that the importance of family may contribute to positive pregnancy experiences in the Hispanic population and, thus, the noted lower adverse neonate outcomes. For instance, a study noted that *familismo*, which indicates strong emphasis of family, was positively associated with a child’s academic performance [27]. However, whether such an association is also relevant for immediate postpartum outcomes needs further exploration. We found no significant association between health insurance status and poor fetal growth, though such an association existed for preterm delivery. Interestingly, mothers with Medicare as the primary payer method demonstrated a significant bivariate association with poor fetal growth and preterm delivery before model adjustment. Whether this was a consequence of small sample size, accounted for by age adjustment, or indicative of another underlying association needs to be explored.

The results of this study should be interpreted within the context of its limitations. Due to the lack of information on post-discharge, follow-up health outcomes cannot be assessed. As with any administrative data, misclassification bias exists, as well as bias from missing data. We also focused on the inland Southern California region, and thus, the results cannot be generalized to the rest of the nation; though areas with similar demographic characteristics may have similar outcomes and thus merit further research. We also cannot assess whether these outcomes are for first births or among those with a history of adverse birth outcomes as the data do not provide patient-level identifiers. Given that our results show the significant association between mental illness and neonate outcomes, future studies could explore whether mothers with history of adverse birth outcomes have a higher prevalence of chronic mental illness. Hospital-level or ZIP code-level characteristics such as poverty level could not be assessed in this study, as the GEE model treated such measures as nuisance variables. Likewise, the use of OSHPD data provides several strengths. Unlike national or state-specific databases that provide a random sample, the data presented in this study are for all observations, and thus, the results directly assess the inland Southern California region. Furthermore, while studies have evaluated the association between maternal mental health and child outcomes, including academic performance, few have addressed immediate birth outcomes. The results of this study add to such a limited body of literature on the negative burden of maternal mental illness on postpartum fetal outcomes. Finally, our results highlight that in order to alleviate the high prevalence of poor maternal–child outcomes in inland Southern California, maternal mental illness should be a key screening and prevention tool. 

## Figures and Tables

**Table 1 ijerph-16-04135-t001:** Prevalence of neonate outcomes among study variables in the South East Desert/Inland Empire region of California, OSHPD 2008–2010. Total sample size *n* = 178, 737 ^†^.

	Poor Fetal Growth	Preterm Delivery
Any Mental Illness	***	***
No	1967 (1.11)	11,811 (6.69)
Yes	58 (2.50)	300 (12.93)
Age Categories	*	***
18–34 years	1726 (1.11)	10,123 (6.52)
35–64 years	299 (1.28)	1988 (8.51)
Primary Payer Type	***	***
Medicare	15 (2.92)	51 (9.94)
Medicaid	981 (1.06)	6215 (6.72)
Private coverage	952 (1.20)	5258 (6.65)
All Other	77 (1.15)	584 (8.73)
Race/Ethnicity	***	***
Non-Hispanic White	565 (1.34)	2902 (6.87)
Hispanic	730 (0.84)	5424 (6.23)
Non-Hispanic Black	117 (1.52)	752 (9.75)
Non-Hispanic Asian/Pacific Islander	42 (1.51)	188 (6.77)
Other	26 (1.34)	123 (6.34)
Charlson–Deyo Index	***	***
0	1884 (1.09)	11,300 (6.53)
1	122 (2.24)	722 (13.24)
2 or more	19 (5.44)	89 (25.50)

OSHPD = California Office of Statewide Health Planning and Development; ^†^ Missing data among study variables may not add up to this number; * *p* < 0.05, ** *p* < 0.01, *** *p* < 0.001.

**Table 2 ijerph-16-04135-t002:** Odds ratio (and 95% CI) of poor fetal growth and preterm delivery in the South East Desert/Inland Empire region of California.

	Poor Fetal Growth	Preterm Delivery
Mental illness present		
Yes	1.79 (1.30, 2.47) **	1.64 (1.41, 1.90) ***
No	1.00	1.00
Age (years)		
18–34	1.00	1.00
35–64	1.13 (0.98, 1.31)	1.37 (1.29, 1.45) ***
Payer type		
Private	1.00	1.00
Medicare	1.57 (0.78, 3.15)	1.08 (0.75, 1.55)
Medicaid	1.06 (0.94, 1.19)	1.06 (1.01, 1.11) *
Other	0.90 (0.68, 1.20)	1.46 (1.32, 1.62) ***
Race/ethnicity		
Hispanic	0.67 (0.60, 0.76) ***	0.91 (0.87, 0.96) **
Non-Hispanic Black	1.16 (0.95, 1.42)	1.37 (1.26, 1.50) ***
Non-Hispanic Asian/Pacific Islander	1.12 (0.82, 1.54)	0.98 (0.84, 1.13)
Non-Hispanic White	1.00	1.00
Other	1.07 (0.72, 1.57)	1.00 (0.83, 1.19)
Charlson Deyo Index		
0	1.00	1.00
1	1.41 (1.10, 1.80) **	1.90 (1.72, 2.09) ***
2 or more	2.81 (1.42, 5.54) **	3.83 (2.85, 5.14) ***

I = Confidence Interval; * *p* < 0.05, ** *p* < 0.01, *** *p* < 0.001.

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
