# Peer review of "Maternal Mental Illness Is Associated with Adverse Neonate Outcomes: An Analysis of Inpatient Data"

_ijerph, 2019, doi:10.3390/ijerph16214135_

Round 1

Reviewer 1 Report

This study aimed to examine the association between maternal mental illness and poor fetal growth and preterm birth using California inpatient data from the OSHPD. Overall, the methods were not clearly described, especially the definition of exposure and outcome and study population. Data were not well presented with limited information provided in the tables. Specific comments are:

1.       The definition of maternal mental illness is unclear. Although it was defined based on the clinical classification software code of 657, more details were needed to describe what mental illness or disorders were included in the assessment.

2.       It is unclear whether the birth outcomes were from first birth only or women could have a history of adverse of birth outcomes, which could significantly affect maternal mental health and birth outcome in the next birth and should be considered in the model.

3.       It is unclear the sample size included in each region in Table 1, and the sample size used for the analyses in Table 2.

4.       Table 2 only shows the adjusted OR; please also provide the descriptive data (number and percentage for each category) to support the result.

5.       Poverty level quartile was defined but was not included in the analyses. This is an important confounder and should be considered in the model.

6.       Authors showed that maternal mental illness was associated with 79% higher odds of poor fetal growth and 71% higher odds of preterm birth, which was not consistent with the data presented in the Table 2 (i.e., 60% higher odds of preterm birth).

7.       Additional analyses using GEE models were not presented and not mentioned anywhere in the manuscript.

8.       The main limitation is that the main analysis was limited to the South East Desert/Inland Empire region, which is a poorer region. Thus, the result cannot be generalised to the state of California (line 190-191). Also, it is hard to justify to conduct analyses on this particular region only, given South Coast region also had a high prevalence of poor fetal grown and preterm birth.   

9.       Based on the results from a single region with a lower SES level, it is insufficient to conclude that maternal mental health status is a key public health issue for birth outcomes.

Author Response

Dear Reviewer,

Thank you for the thorough feedback. We re-analyzed the data as requested in key areas, added more data points, analysis, and also updated the manuscript itself. We appreciate the feedback that has now made this paper strong. Our responses are added below in bold for ease of finding.

This study aimed to examine the association between maternal mental illness and poor fetal growth and preterm birth using California inpatient data from the OSHPD. Overall, the methods were not clearly described, especially the definition of exposure and outcome and study population. Data were not well presented with limited information provided in the tables. Specific comments are:

The definition of maternal mental illness is unclear. Although it was defined based on the clinical classification software code of 657, more details were needed to describe what mental illness or disorders were included in the assessment.

Response: The specific disorders have been updated in the manuscript and a reference to AHRQ HCUP CCS code definitions has been added for clarity. Thank you for your suggestions.

It is unclear whether the birth outcomes were from first birth only or women could have a history of adverse of birth outcomes, which could significantly affect maternal mental health and birth outcome in the next birth and should be considered in the model.

Response: The definition for birth outcomes has been clarified, which excludes abnormal (e.g., molar or ectopic pregnancies) or abortive pregnancy outcomes. History of other births or adverse birth outcomes is not possible using this dataset, as patients cannot be linked using this public use data set. This suggestion has now been included in the discussion section as well.

It is unclear the sample size included in each region in Table 1, and the sample size used for the analyses in Table 2.

Response: The updated table 1 now has sample sizes.

Table 2 only shows the adjusted OR; please also provide the descriptive data (number and percentage for each category) to support the result.

Response: This table has now been added as the updated table 1. 

Poverty level quartile was defined but was not included in the analyses. This is an important confounder and should be considered in the model.

Response: Poverty level quartile was defined as a ZIP code-level variable and included as part of the subject-effect clusters within the GEE model, along with hospital ID. This has now taken into account the effect of poverty level quartile and all other ZIP code level characteristics as a nuisance variable, but as a result we are not able to account for this at the individual level. This was included as a potential limitation.

Authors showed that maternal mental illness was associated with 79% higher odds of poor fetal growth and 71% higher odds of preterm birth, which was not consistent with the data presented in the Table 2 (i.e., 60% higher odds of preterm birth).

Response: The tables have been updated.

Additional analyses using GEE models were not presented and not mentioned anywhere in the manuscript.

Response: As the manuscript now focuses on the South East Desert/Inland Empire region, these GEE models are not needed. Thank you for bringing this to our attention.

The main limitation is that the main analysis was limited to the South East Desert/Inland Empire region, which is a poorer region. Thus, the result cannot be generalized to the state of California (line 190-191). Also, it is hard to justify to conduct analyses on this particular region only, given South Coast region also had a high prevalence of poor fetal grown and preterm birth.   

Response: Thank you for the feedback, we updated the section to focus on a region as well as updated the limitations section.

Based on the results from a single region with a lower SES level, it is insufficient to conclude that maternal mental health status is a key public health issue for birth outcomes.

Response: That was such a good point that’s why we decided to refocus on this region alone due to the substantial socioeconomic burden as well as history of poor maternal child outcome. Thus, our study further adds whether mental illness is a key factor as well. We have also added this to the limitation that we cannot generalize to the entire state or nation.

Reviewer 2 Report

In the current study, Lopez et al. have investigated the inter-relationship between mothers with poor mental health and its effect on fetal health. The study provides insights into how maternal mental health serves as risk factor for fetal growth and survival outcomes. Evidence based studies are important for physicians to come with appropriate interventions and provide treatment considerations for improving maternal and neonatal health. Overall, this is a well-designed study with appropriate statistical analyses. However, there are a few concerns that the authors need to address before the manuscript can be considered for publication.

1.     The authors have focused on the prevalence of maternal mental illness and neonate outcomes by geographic region. It is important to mention the sample size in both table 1 and 2.

2.     The study involved participants of age 18-45 years old. However, the data in table 2 shows age group of 15 onwards. Please explain the discrepancy. Also, the data for age group 40 and above is missing in table 2.

3.     The authors report 60% higher odds of preterm birth in table 2 but state 71% in their results section. Please explain the discrepancy.

4.     Do the authors consider poor fetal growth as IUGR? Is it possible to subcategorize poor fetal growth category further? What about the comparison of maternal mental illness with low birth weight outcomes?

5.     It is interesting to note that increased odds of preterm birth were associated with age, ethnicity and other co-morbidities. The authors need to discuss in detail as to why no association was seen in terms of poor fetal growth in this sub-population. The authors need to comment on low OR observed in Hispanic group in terms of poor fetal growth.

6.     The mean gestational age of the mothers at the time of delivery needs to be reported. How many of preterm pregnancies were spontaneous and/or induced?

7.     In line 144-145, the authors state the increased burden shared among minorities and those of low socioeconomic status. As no significant changes were observed in other minority groups (Hispanics and Asians), the authors need to limit their findings to African Americans and not state minorities in general.

8.     The authors need to include the demographic and clinical characteristics of the participants in detail. Demographic (with respect to socio-economic status, employment, education, marital status), obstetric (like gestational age, singleton pregnancy), medical and psychiatric characteristics etc. Also, the authors need to subcategorize mental illness with respect to anxiety, depression, psychotic, mood, eating disorders etc. Other factors relating to lifestyle (like smoking and drinking) need to be provided as they could be confounding factors in terms of preterm and poor fetal growth.

9.     In general, the authors need to improve the writing skills in the manuscript. Also, consider using et al. when you are citing more than two authors and omitting et al. when citing studies having two authors.

Author Response

Dear Reviewer,

Thank you for the thorough feedback. We re-analyzed the data as requested in key areas, added more data points, analysis, and also updated the manuscript itself. We appreciate the feedback that has now made this paper strong. Our responses are added below in bold for ease of finding.

In the current study, Lopez et al. have investigated the inter-relationship between mothers with poor mental health and its effect on fetal health. The study provides insights into how maternal mental health serves as risk factor for fetal growth and survival outcomes. Evidence based studies are important for physicians to come with appropriate interventions and provide treatment considerations for improving maternal and neonatal health. Overall, this is a well-designed study with appropriate statistical analyses. However, there are a few concerns that the authors need to address before the manuscript can be considered for publication.

The authors have focused on the prevalence of maternal mental illness and neonate outcomes by geographic region. It is important to mention the sample size in both table 1 and 2.

Response: Thank you, however since the paper has been refocused to only a single region per reviewer commentary, this is no longer necessary. We appreciate your input.

The study involved participants of age 18-45 years old. However, the data in table 2 shows age group of 15 onwards. Please explain the discrepancy. Also, the data for age group 40 and above is missing in table 2.

Response: This was due to a mislabel from the data analysis tool and has been fixed.

The authors report 60% higher odds of preterm birth in table 2 but state 71% in their results section. Please explain the discrepancy.

Response: We updated all the tables and double-checked with narrative. This should be now fixed.

Do the authors consider poor fetal growth as IUGR? Is it possible to subcategorize poor fetal growth category further? What about the comparison of maternal mental illness with low birth weight outcomes? 

Response: Not able to subcategorize due to low sample size, as the outcome is based on the two codes

50 Poor fetal growth, affecting management of mother, unspecified as to episode of care or not applicable 51 Poor fetal growth, affecting management of mother, delivered, with or without mention of antepartum condition there doesn’t seem to be sufficient information by splitting the two codes.

It is interesting to note that increased odds of preterm birth were associated with age, ethnicity and other co-morbidities. The authors need to discuss in detail as to why no association was seen in terms of poor fetal growth in this sub-population. The authors need to comment on low OR observed in Hispanic group in terms of poor fetal growth.

Response: We have added feasible explanations for the lack of association on some variables in our discussion; though we acknowledge the need for future exploratory research in this area, especially by racial/ethnic group.

The mean gestational age of the mothers at the time of delivery needs to be reported. How many of preterm pregnancies were spontaneous and/or induced?

Response: Mean age was not shown as specific age in years had substantial missing data, and the categorized version of age which had fewer missing data was used. Spontaneous and/or induced pregnancies were not assessed in this study.

In line 144-145, the authors state the increased burden shared among minorities and those of low socioeconomic status. As no significant changes were observed in other minority groups (Hispanics and Asians), the authors need to limit their findings to African Americans and not state minorities in general.

Response: We have updated this accordingly.

The authors need to include the demographic and clinical characteristics of the participants in detail. Demographic (with respect to socio-economic status, employment, education, marital status), obstetric (like gestational age, singleton pregnancy), medical and psychiatric characteristics etc. Also, the authors need to subcategorize mental illness with respect to anxiety, depression, psychotic, mood, eating disorders etc. Other factors relating to lifestyle (like smoking and drinking) need to be provided as they could be confounding factors in terms of preterm and poor fetal growth.

Response: Further detail regarding the demographic and clinical characteristics has now been provided. Due to the small sample sizes, we could not do a sub-analysis of specific mental illnesses in this category.

In general, the authors need to improve the writing skills in the manuscript. Also, consider using et al. when you are citing more than two authors and omitting et al. when citing studies having two authors.

Response: We have updated our manuscript and addressed any grammatical concerns.

Round 2

Reviewer 1 Report

The methods, results, and tables have been improved. There are minor comments to improve the manuscript further.

In the Data Variables section, please double check the CCS code for mood disorder and impulse control disorders. Are they both coded as 657? In the Data Variables section, age ranged from 18 to 45 years; however, the data presented in Table 1 & 2 and Results section (1st paragraph) were categorised as 18-34 and 35-64 years. Please clarify the age range included for the analysis. For the race/ethnicity, please consider removing “non-Hispanic” for the White, Black and Asian categories, unless there is a specific reason. If agree, please update it throughout. Line 157 in the 2nd paragraph of the Results section, increased odds of preterm delivery associated with increasing co-morbidities has been mentioned in the previous sentence. Please do not repeat that result.   Please provide more details on the differences between Medicare and Medicaid in the Methods. In Table 2, please revise “Medical” to “Medicaid” for consistency. Women with Medicare seemed to have a higher prevalence of poor fetal growth and preterm delivery (Table 1), though there was no significant association after adjustment, partly due to small sample of this group and insufficient power (wide CI)? Line 165-167 of Discussion, the authors concluded two key findings of the study. However, the increased burden among Black mothers was mainly for preterm delivery, but not for poor fetal growth, compared to White mothers. The other key finding is that maternal co-morbidities was also a key factor for adverse birth outcomes. Line 172-173 of Discussion, it is not clear that increasing units on a depression scale (I assumed it means more depressed) was associated with higher or lower birth weight? There are several typos – please proofread. Two examples listed below: Line 163: “We specifically those these variables as they have been….” – specifically chose? Line 216: “a higher prevalence or chronic mental illness..” – of.

Author Response

In the Data Variables section, please double check the CCS code for mood disorder and impulse control disorders. Are they both coded as 657? 

Response: Thank you for bringing this to our attention. This has been updated. Impulse control disorders code was 656.

In the Data Variables section, age ranged from 18 to 45 years; however, the data presented in Table 1 & 2 and Results section (1st paragraph) were categorised as 18-34 and 35-64 years. 

This should have been ranged 18 to 64 years and has been updated. We appreciate your feedback.

Please clarify the age range included for the analysis. For the race/ethnicity, please consider removing “non-Hispanic” for the White, Black and Asian categories, unless there is a specific reason. If agree, please update it throughout. 

Response: Age has been clarified to 18 to 64 years. However, the way OSPHD gave the race/ethnic data, Hispanic was separate from race, and as such, we had to create independent variables that were Hispanic, versus non-Hispanic others (Black, White, etc.).

Line 157 in the 2nd paragraph of the Results section, increased odds of preterm delivery associated with increasing co-morbidities has been mentioned in the previous sentence. Please do not repeat that result.  

Response: This was noted and corrected.

Please provide more details on the difference between Medicare and Medicaid in the Methods.
Response: We have added this section in the methods.

In Table 2, please revise “Medical” to “Medicaid” for consistency. 

Response: This has been revised.

Women with Medicare seemed to have a higher prevalence of poor fetal growth and preterm delivery (Table 1), though there was no significant association after adjustment, partly due to small sample of this group and insufficient power (wide CI)? 

Response: Yes, this is a good point and has been added to the discussion. We appreciate your feedback.

Line 165-167 of Discussion, the authors concluded two key findings of the study. However, the increased burden among Black mothers was mainly for preterm delivery, but not for poor fetal growth, compared to White mothers. The other key finding is that maternal co-morbidities was also a key factor for adverse birth outcomes. 

Response: Thank you, this has been clarified in the discussion.

Line 172-173 of Discussion, it is not clear that increasing units on a depression scale (I assumed it means more depressed) was associated with higher or lower birth weight? 
Response: We have updated that section to state that increasing depressive symptoms was associated with lower birth weight. 

There are several typos – please proofread. Two examples listed below: Line 163: “We specifically those these variables as they have been….” – specifically chose? 

Response: This has been noted and corrected.

Line 216: “a higher prevalence or chronic mental illness..” – of. 

Response: This has been noted and corrected.